# Social media shines light on the "hidden" impact of nighttime guided-gigging charters on Texas' Southern Flounder fishery: A stab in the dark

Quentin A. Hall[1]☯, Daniel M. Coffey[1]☯, Matthew K. Streich[1]☯, Mark R. Fisher[2], Gregory W. Stunz[1]*

1 The Center for Sportfish Science and Conservation, Harte Research Institute for Gulf of Mexico Studies, Corpus Christi, TX, United States of America, 2 Texas Parks and Wildlife, Coastal Fisheries Division, Rockport, TX, United States of America

☯ These authors contributed equally to this work.

* greg.stunz@tamucc.edu

**Data Availability Statement:** As data were gathered exclusively through Facebook and were publicly available, no privacy issues were violated. User content was public, and we abided by the

## Abstract

Southern Flounder (*Paralichthys lethostigma*) populations are declining in the Gulf of Mexico basin. This is particularly true in Texas, where this unique and culturally important fishery has been in decline since the 1980s despite increasingly stringent regulatory measures. Current angler-intercept creel surveys used to estimate recreational flounder harvest levels are conducted during daylight hours and do not account for the high levels of nighttime flounder gigging (spearing) activity, a popular and efficient harvest method for this fishery. There are legitimate scientific and logistical concerns that have prevented the use of wide-spread nighttime creel surveys to monitor the flounder gigging fishery in the past, however this has made accurate catch and effort estimates difficult to obtain. Given the concern about this economically important fishery's status, we adopted a unique approach utilizing social media to provide unprecedented information into this fishery's impact during periods that are not traditionally monitored. Specifically, we reconstructed seasonal flounder harvest and effort metrics stemming from the nighttime recreational guided flounder gigging sector over 2.6 years using guided flounder gigging charter photo archives publicly available through Facebook. These metrics show large average client party sizes, large trip harvests, and near-perfect bag limit efficiencies. Temporal trends indicated peak recreational guided flounder gigging effort and harvest occurs during the summer months, a time not traditionally associated with flounder gigging. The addition of nighttime guided-gigging recreational harvest estimates from this study to traditional daytime harvest estimates and commercial harvest estimates resulted in total annual harvest estimates nearly two times greater than current estimates. Overall, this study demonstrates the high pressure guided-gigging charters are placing on Texas' flounder fishery and illustrates the critical need for additional information on the nighttime recreational flounder fishery for both guided and private gigging anglers. Moreover, our results also demonstrate the usefulness of mining social media platforms to capture catch and effort data that are otherwise unavailable.

terms, conditions, and restrictions set by Facebook and PLOS ONE's privacy policies. Due to legal requirements ensuring anonymity of data sources, an anonymized version of the data underlying the results of this study are archived and freely available from the Gulf of Mexico Research Initiative Information and Data Cooperative (GRIIDC; https://data.gulfresearchinitiative.org/; doi: 10.7266/NBHGK8RA).

**Funding:** The authors received no specific funding for this work.

**Competing interests:** The authors have declared that no competing interests exist.

## 1—Introduction

Southern Flounder (*Paralichthys lethostigma*) are an estuarine-dependent species best known for their late fall spawning migrations (October-December) and are among the most targeted fish in Texas marine waters [1–4]. Historically, Southern Flounder have supported an important multimillion-dollar commercial and recreational fishery along the Texas coast and the broader Gulf of Mexico (Gulf) [5, 6]; however, the Gulf's, especially Texas', Southern Flounder populations have been in steady decline since the 1980s prompting numerous management revisions aimed at reducing harvest and effort in the fishery [7–9]. Despite these regulatory changes, Southern Flounder have continued to experience an overall decline [8, 10, 11].

Texas' Southern Flounder populations support economically important commercial and recreational fisheries in which gigging at night (spearing by lighted means) is the predominant harvesting method [12]. Flounder gigging is also culturally important as a popular and time-honored tradition along the Texas coast. Historically, gigging was a nighttime activity most often practiced during the fall migration season when Southern Flounder concentrate in tidal passes while migrating offshore to spawn [13]. Traditional gigging equipment most often consists of a multi-pronged spear (gig), hand-held light or lantern, and requires participants to wade shallow water in search of fish [14]. However, rapid advances in shallow water boat capabilities and illumination technology have led to the development of specialized flounder gigging boats that are very common and extremely efficient in this fishery. These boats often utilize halogen or LED light systems powered by generators or batteries. Furthermore, these boats can navigate extremely shallow water using air-cooled engines powering small airplane propellers known as "push fans." The new technology on these specialized vessels allows large groups of anglers to cover miles of shallow water in search of flounder over an incredibly short period of time. The ability to efficiently search large areas has eliminated the need to wait for flounder to congregate during the fall spawning migration, resulting in increased fishing pressure on flounder year-round. These technological advancements have also given rise to numerous for-hire guided-gigging charters in which charter captains take paying clients to harvest Southern Flounder. Guided-gigging charters perceive the Texas flounder population as healthy as was observed at numerous Texas Parks and Wildlife Department (TPWD) flounder regulation scoping meetings in 2019 (Q. A. Hall, pers. obs). This contrasts with scientific data from TPWD's fishery-independent surveys demonstrating long-term declines. These charters have led to controversy within the recreational fishery and present a unique challenge to fisheries management. The decline of Southern Flounder in Texas is commonly attributed to poor recruitment related to changing climate patterns and adult overharvest, yet the relative importance of these two factors remains unresolved. TPWD fishery-independent surveys have revealed a long-term decline of Southern Flounder recruitment in Texas bays coinciding with coastwide increases in water temperature during the spawning season (November-February) [8, 11]. Growth and survival of newly hatched Southern Flounder larvae in Texas waters is greatest at 18°C and highly sensitive to relatively small changes in temperature (± 2°C) [15]. Southern Flounder also have environmental sex determination with masculinization at warmer temperatures [9, 16, 17]. Because female Southern Flounder attain larger sizes than males (and consequently dominate fishery harvest), temperature-driven changes in sex ratios would also alter the size structure and biomass of the population [18]. Thus, increases in water temperature could have substantial negative effects on the recruitment, population size and demographics of Southern Flounder [18, 19]. For example, Texas' Southern Flounder populations continue to decline in the Sabine Lake (north) and Lower Laguna Madre (south) bay systems despite more limited fishing effort (perceived) relative to the central Texas coast [20].

Although poor recruitment and changing sex ratios have likely contributed to reduced population abundance, Froeschke et al. [11], using data from TPWD fishery-independent surveys (1975–2008), determined that adult, (≥290 mm total length (TL)) [10, 13], Southern Flounder are declining twice as fast as juveniles. This indicates that the larger decline seen in the adult population may be less related to recruitment limitation and could be a result of overfishing and incidental bycatch in commercial shrimp trawl fisheries. However, historical Southern Flounder bycatch in the Texas commercial shrimp trawl fishery was primarily composed of juveniles [21–23] and bycatch reduction device requirements in Texas state and Gulf of Mexico federal waters since 1998 have become more efficient at excluding larger individuals [6]. In 2002, Texas implemented a commercial limited entry and buy-back program of shrimp vessels, which reduced overall bycatch by at least 80% [23]. In comparison, Froeschke et al. [11] reported the overall mean size of adult Southern Flounder captured in fishery-independent surveys was 360 mm TL, which is remarkably close to the 14-inch (356 mm TL) minimum size limit established in 1996 (recently increased to 15-inches [381 mm TL] in 2020) for recreational and commercial fisheries. Thus, the reported decline in adult Southern Flounder abundance at size classes exceeding the minimum size limit is likely attributed more to targeted fishing pressure than bycatch; though, increased natural mortality of larger adults is also a contributing factor [11]. Given this information, additional research is critically needed to determine factors contributing to overfishing and promote sustainable management strategies for Southern Flounder in Texas [19].

In Texas, TPWD is responsible for monitoring Southern Flounder harvest and effort by the commercial and recreational sectors of the fishery. While commercial harvest reporting includes all fishing activity (i.e., day/night, gear type), routine harvest monitoring of the recreational sector through the Texas Marine Sport-Harvest Monitoring Program employs angler-intercept surveys at boat access sites only during daytime hours (1000 to 1800 hours) [20, 24, 25]. Past studies have revealed that significant numbers of Southern Flounder are harvested at night by gig anglers that are not intercepted during routine daytime surveys, which could result in severely underestimating harvest patterns of Southern Flounder [13, 14, 26]. Additionally, Olsen and Wagner [14], discuss that limited nighttime creel surveys conducted by TPWD had much lower survey efficiencies than traditional daytime creel surveys. This further complicates the development of effective nighttime recreational harvest and effort monitoring strategies, leading to recreational gigging activities occurring almost completely unmonitored.

Previous TPWD estimates of nighttime flounder gigging effort indicate recreational anglers account for the majority of effort (83–89%) and 53% of landings compared to commercial operations [20]. However, the lack of replication (sampling only a few months in a single year) and data collection date (1991 and 2007) of these studies prevent conclusions about the annual seasonality and magnitude of nighttime flounder gigging effort and harvest, highlighting the critical need for updated information for this nighttime fishery. Most guided-gigging charters begin operating at 1900 hours or later and operate year-round. Thus, nighttime recreational gigging harvest remains unaccounted for in recreational harvest estimates [14, 20]. Given the concern about this economically important fishery's status and the lack of nighttime recreational harvest data, analyses based on non-traditional data sources may provide unprecedented information into this fishery sector's impact.

Recently, novel approaches are increasingly used to extract publicly available, open-source information from social media platforms to reveal new insights into the study of particular species [27–30] and address conservation issues [31, 32]. Many guided-gigging charters advertise by sharing photos of their client's catch at the end of a trip on social media, particularly on Facebook. Some of these photo archives are extensive and include several years' worth of catch photos containing harvest data, client party size, and temporal information. Thus, social

media provides a unique opportunity to quantify nighttime guided-gigging harvest and effort metrics during periods that are not traditionally monitored. Given the large data gaps described above, we used guided-gigging charter photo archives on Facebook to provide a novel characterization of the guided-gigging component of the nighttime recreational flounder fishery in Texas. The specific objectives of this study were to: (1). Estimate seasonal flounder harvest and effort metrics associated with guided flounder gigging charters within Texas waters; (2). Compare metrics from this study to traditional flounder harvest estimates generated by the TPWD; (3). Provide managers with potential approaches to curb Southern Flounder population declines within the state based on this new information.

## 2—Methods

### Guided-gigging charter selection

There are no official records detailing how many licensed charter captains operate guided-gigging charters in Texas. As such, an internet search using all combinations of the terms "gigging", "guide", "service", "flounder", and "Texas" covering fishing forums, outdoor classified boards, social media platforms, and general search engines was conducted in April 2020 to determine an estimated number of licensed captains offering guided-gigging charters in Texas. From the 21 guided-gigging businesses located, a subset of three was selected to obtain data for this study. Each was selected given that their businesses' Facebook pages housed photo archives containing hundreds of photos documenting flounder gigging trips dating back to mid-April 2017. All three guided-gigging charters operate gigging boats along the central Texas coast, as this high-effort geographic area is of particular interest to managers [20]. It is worth noting that none of these charters operate in the Galveston Bay system, which is frequently associated with high levels of recreational flounder effort (albeit historically rod and reel) [33]. With limited available data, we assumed these charter operations were representative of average guided-gigging businesses. Photo archives from May 1, 2017, to December 31, 2019, were selected as this period contained the longest continuous interval in which extensive photo records could be found for each of the three selected charters and provided appropriate replication to determine seasonal pressure, another area of particular interest for managers [20]. Photo data were quantified in April 2020.

### Data processing

Each photo was assumed to represent an individual charter trip (usually confirmed based on photo description or client party size), and all fish pictured were identified to the lowest taxonomic level possible and counted. We did not attempt to distinguish between Southern Flounder and Gulf Flounder (*P. albigutta*) as TPWD manages both species collectively [34], and Southern Flounder represent 95% of flounder harvested within Texas waters [1, 11]. Additional species that can be legally taken by gigging (hereafter referred to as ancillary species) include Black Drum (*Pogonias cromis*), Sheepshead (*Archosargus probatocephalus*), Alligator Gar (*Atractosteus spatula*), Florida Pompano (*Trachinotus carolinus*), and stingrays (*Dasyatis* spp.). The vertical bars on juvenile Black Drum may be morphologically similar to those on Sheepshead. Consequently, individual Black Drum and Sheepshead that could not be easily distinguished due to poor image quality were collectively referred to as "unidentified fish." Blue Crabs (*Callinectes sapidus*) were also harvested, but they were not examined in this study as it proved too difficult to enumerate them in the photos. The number of clients visible within the photo was also recorded. Trips were assumed to have occurred on the day each picture was posted unless otherwise specified in the photo description. It was also assumed that each guide posted photos of every trip they conducted each year. While unlikely, this assumption is biased

towards conservative harvest estimates. Charter trip photo duplicates were identifiable by pictured clients, boat ramp locations, and fish counts. In addition, several guides also posted commercial flounder gigging photos. These photos, along with charter trip duplicates, were discarded from subsequent analyses. For each photo, two independent readers made blind counts of each species present and the number of clients. When counts differed, the photo was jointly examined to reach a consensus. Several photos were posted on closed season dates, however associated comments specified these trips had taken place earlier during the open season. All data used in this study were publicly available through social media and remain anonymous with no collection of personal information.

The Texas flounder fishing regulations that were in place during the study period stipulated that no flounder could be taken by gigging from November 1–30, and only two flounder per person per day could be taken by any means from December 1–14 (hereafter the "restricted season"). A total of five flounder per person per day could be taken by any means for the remainder of the year (hereafter the "regular season") [34]. During this study, the regular season ran 321 legal fishing days each calendar year, exclusive of the 30-day closure in November and 14-day restricted season during the beginning of December. Because this study began on May 1, 2017, only 201 legal fishing days were surveyed during the regular season for that year for a grand total of 843 legal fishing days from 2017–2019. A grand total of 42 legal fishing days were surveyed for the restricted season from 2017–2019. Data were analyzed separately according to their respective season.

## Trip metrics

As individual trip length (i.e., number of hours spent gigging) was unknown, fishing effort was reported as the estimated number of guided fishing trips. The average number of trips per season was calculated for each guide and then averaged across all guides. In addition, the average number of trips per legal fishing day was calculated to determine what percentage of legal days guides chartered a least one trip in a given season. These metrics were divided by the number of legal fishing days within each season to determine the average number of trips taken by each guide on any legal fishing day. There were several instances in which the number of flounder exceeded the daily bag limit for a guided fishing party based on the number of clients pictured. We firmly believe this was not due to illegal activity, but rather that not all clients were pictured. This assumption was further supported by multiple Facebook post comments explaining that certain clients did not want to be pictured. Other comments explained that some trips were one flounder short of a limit because the guide miscounted and thought the limit had already been reached. Given this evidence, we were confident each of the surveyed guides operated within the law. Thus, guided fishing party size could be estimated by the total number of observed flounder. Accordingly, the number of observed flounder for each trip was divided by the daily bag limit for the corresponding season to estimate party size for trips in which the total number of flounder exceeded the legal bag limit based on the number of clients pictured. The average number of clients per trip per season was calculated for each guide and then averaged across all guides.

## Harvest metrics

The average number of flounder harvested per trip each season was calculated for each guide and then averaged across all guides. The observed number of flounder harvested was divided by the bag limit for a guided fishing party (based on party size and seasonal regulations) to determine the average percentage of a trip limit harvested on each trip. The percentage of trips harvesting ancillary species was calculated for each season in addition to the average number

of ancillary fish harvested per trip across all guides. Following Ajemian et al. [35], a monthly catch proportion was calculated for each species (the number of individuals of a species harvested in a single month divided by the total number of individuals harvested in that same month), excluding flounder. For each guide, monthly catch proportions were averaged across years, and the resulting values were averaged across all three guides.

## Temporal trends

Temporal trends in the number of trips, flounder harvested per trip, total flounder harvested, and clients per trip, were assessed by calculating average monthly values per year for each guide and then averaging across years, accounting for the lack of data for January-April 2017. These monthly average values were pooled to calculate averages (± standard error) across all guides.

## Traditional flounder harvest estimate comparison

Daytime recreational and commercial flounder harvest data from 2017–2019 were obtained through TPWD's Marine Sport-Harvest Monitoring Program and commercial harvest reporting system, respectively. Through the Marine Sport-Harvest Monitoring Program, harvest and effort are estimated from angler-intercept surveys conducted during daytime hours (1000–1800 hours) throughout the year in each of the eight major Texas bay systems (see Green [33] for additional details). Commercial harvest was estimated based on trip ticket reporting and was provided in pounds total weight per year [36]. To estimate the number of fish harvested by the commercial fleet each year, total weight values were divided by 2.0 pounds which is the average weight for flounder harvested in Texas' commercial flounder fishery [8]. Collectively, these datasets estimate flounder total annual harvest for daytime private and guided recreational harvest as well as commercial harvest. To compare these estimates with annual nighttime guided recreational gigging harvest, we calculated mean (± standard error) annual guide flounder harvest $(\hat{H})$ by averaging total flounder harvest across all three guides. Total annual nighttime guided recreational harvest (± standard error) was then estimated by multiplying the mean annual guide flounder harvest $(\hat{H})$ by the number of gigging guides identified through our internet search following [37]. It is uncertain whether all of the statewide guided-gigging charters harvest at a similar average annual rate to the three guided-gigging charters analyzed in this study. To address this potential sampling bias, we conducted a sensitivity analysis to determine how different numbers of statewide guided-gigging charters affect the total annual nighttime guided recreation harvest compared to daytime private and guided recreational harvest and commercial harvest.

## 3—Results

### Trip metrics

A total of 1618 photos were read, representing 1385 unique guided-gigging trips surveyed during the 2.6 regular seasons (excluding January-April 2017) included in the study period and an additional 64 trips surveyed during the three restricted seasons. Guides averaged 153.9 (SE = 29.4) trips each regular season (54% of legal fishing days) and 7.1 (SE = 2.2) trips each restricted season (52% of legal fishing days), or approximately one trip every two days (Table 1). The average party size per trip was estimated to be 3.9 (SE = 0.1) clients during the regular season and 4.3 (SE = 0.3) clients during the restricted season (Table 1).

Summary of Nighttime Guided Recreational Flounder Gigging Trip and Harvest Data by Season accounting for lack of January-April 2017 data. The Texas flounder fishing regulations

**Table 1. Trip and harvest summary.**

| Summary of Nighttime Guided Recreational Flounder Gigging Trip and Harvest Data by Season | | |
|---|---|---|
| | Regular Season | Restricted Season |
| Total Number of Trips Surveyed | 1385 | 64.0 |
| Avg. Number Trips/season/guide | 153.89 | 7.1 |
| Avg. Number Trips/legal fishing day/guide | 0.54 | 0.52 |
| Avg. Number Clients/trip/guide | 3.9 | 4.3 |
| Total Number Harvested Flounder Observed | 24899 | 506 |
| Avg. Number Flounder Harvested/trip/guide | 18.2 | 8.5 |
| Avg. Total Number Flounder Harvested/season/guide | 2766.6 | 56.2 |
| Avg. % of allowable trip limit harvested/trip | 95.2% | 98.6% |
| Percentage of trips harvesting ancillary species | 61.3% | 76.6% |
| Avg. number of ancillary fish harvested/trip/guide | 2.1 | 5.7 |

that were in place during the study period stipulated that no flounder could be taken by gigging from November 1–30, and only two flounder per person per day could be taken by any means from December 1–14 (restricted season). A total of five flounder per person per day could be taken by any means for the remainder of the year (regular season).

## Harvest metrics

Among the three guides surveyed, a total of 24,899 and 506 flounder were harvested during regular and restricted seasons, respectively, during the study period. Each guide harvested an average of 18.2 flounder per trip and reached their limit 95.2% of trips during the regular season (Table 1). In contrast, an average of 8.5 flounder were harvested per trip and party limits were reached 98.6% of trips during the restricted season. Each year, an average of 2766.6 flounder were harvested during the regular season, and an average of 56.2 flounder were harvested during the restricted season by each guided charter. Flounder accounted for 67.7% (January) to 93.8% (June) of average monthly catch proportions. In addition to flounder, 61.3% of regular-season trips harvested ancillary species, with an average of 2.1 fish per trip (Table 1). In contrast, 76.6% of restricted-season trips harvested ancillary species, with an average of 5.7 fish per trip (Table 1). Average monthly catch proportions (excluding flounder) indicate that Sheepshead and Black Drum comprised the majority of harvested ancillary species (Fig 1). These values ranged from 32.1% (January) to 77.9% (May) for Sheepshead and 11.9% (May) to 67.9% (January) for Black Drum. Unidentified species (comprised of indistinguishable Sheepshead and Black Drum) accounted for an additional 0% (January-February) to 10.2% (May) of average monthly catch proportions of ancillary species. Alligator Gar, Florida Pompano, and stingray harvest infrequently occurred (0% to 2.4% of average monthly catch) and only during particular months of the year.

## Temporal trends

Monthly effort for guided-gigging charters was highest from March (17.8 trips/guide) to August (17.1 trips/guide), with a peak in July (26.6 trips/guide) (Fig 2A). A secondary peak in average trip numbers occurred in October (19.2 trips/guide), coinciding with the start of the flounder spawning migration and the last month before the November gigging closure (Fig 2A). The average number of clients per trip peaked in the restricted season (December 1–14) at 4.32 clients/trip. The average number of clients per trip remained steady between March (3.8 clients/trip) and July (4.3 clients/trip) (Fig 2B). The average number of flounder harvested

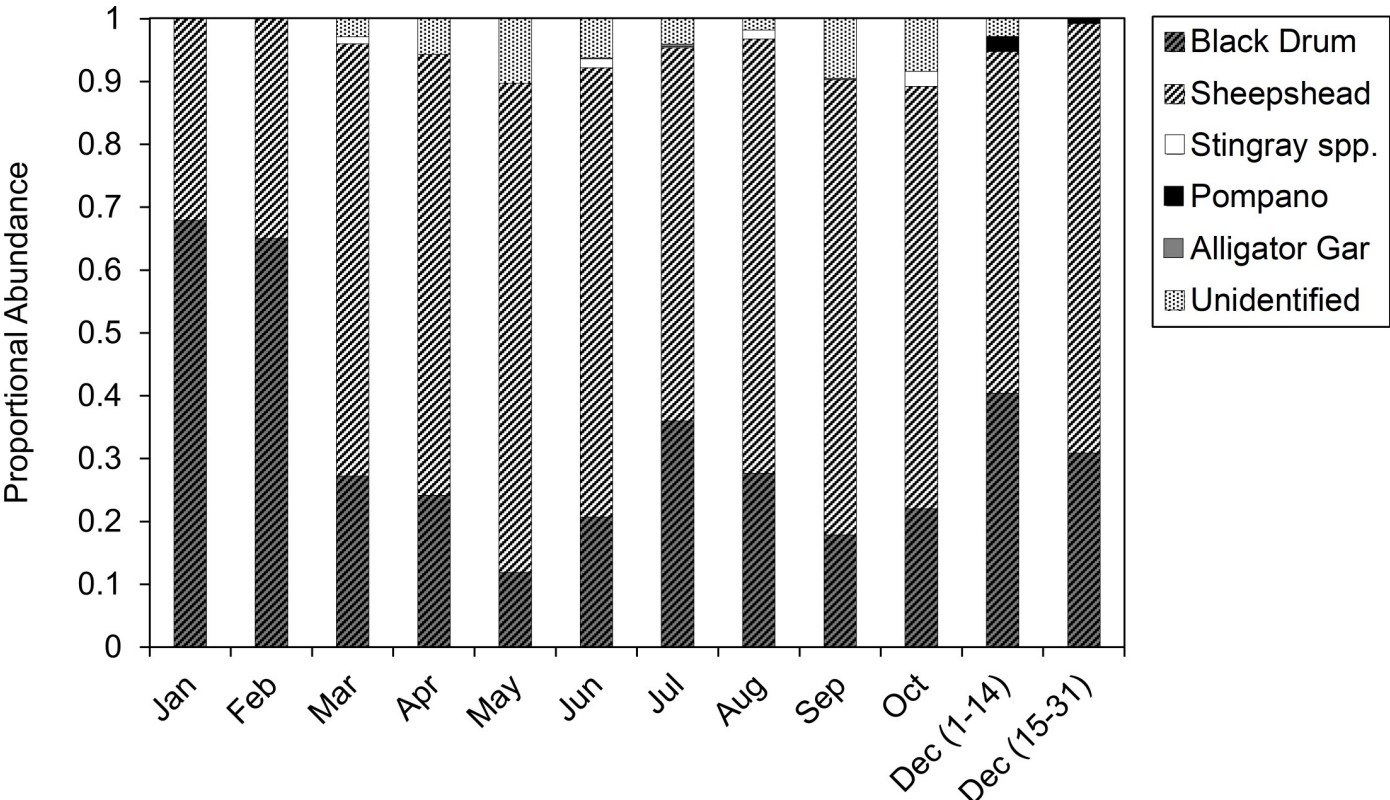

**Fig 1. Ancillary species catch proportions (excluding flounder).** Average monthly catch proportions (excluding flounder) of ancillary species (coded by color and pattern) for guided-gigging charters over the study period. Gigging is not allowed in November.

per trip peaked in July (20.2 flounder/trip); though, trip harvest levels remained remarkably consistent from March to July, ranging between 18.0 to 20.2 flounder/trip (Fig 3A). Moreover, we note the smaller standard errors indicate a high level of consistency with which guided-gigging charters harvest flounder within these months. The highest average monthly harvest occurred between March (320.2 flounder/guide) and August (310.8 flounder/guide), with a peak in July (534.3 flounder/guide) (Fig 3B).

## Traditional flounder harvest estimate comparison

We identified a total of 21 licensed charter captains that operate guided-gigging charters targeting Southern Flounder in Texas waters. The sensitivity analysis demonstrated that if one additional nighttime guided-gigging charter (19% of identified charters) operates at the same average annual rate as the three charters we surveyed in this study, the annual nighttime guided-gigging Southern Flounder harvest would exceed the total annual daytime guided recreational harvest in 2017–2019 (Fig 4). Notably, the three charters we surveyed in this study alone exceed the total annual daytime guided recreational harvest in 2018 by 5,595 (228%) Southern Flounder. Similarly, if one additional nighttime guided-gigging charter operates at the same average annual rate as the three charters we surveyed in this study, the annual nighttime guided-gigging Southern Flounder harvest would exceed the total annual commercial harvest in 2018 and 2019. The total annual commercial Southern Flounder harvest was highest in 2017 (19,922 flounder) and would require seven additional nighttime guided-gigging charters (48% of identified charters) to operate at the same average annual rate as the three charters

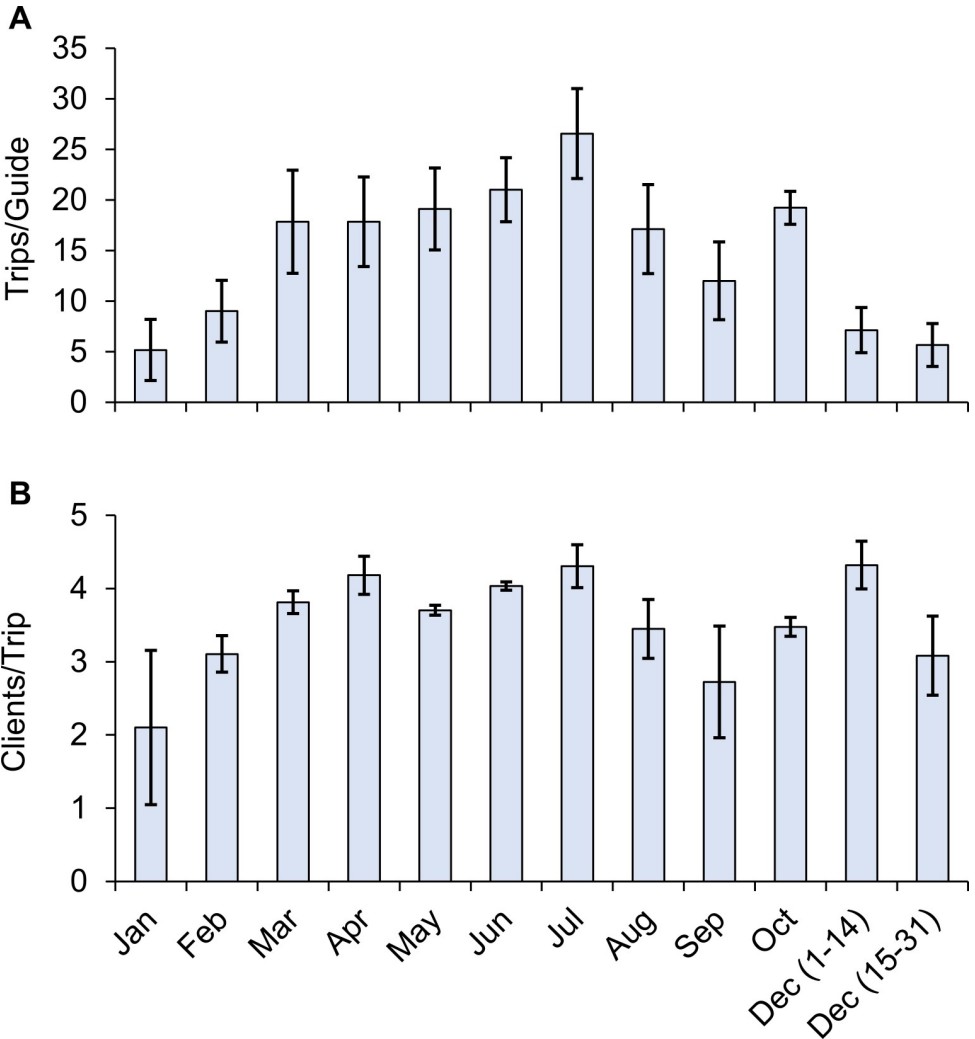

**Fig 2. Monthly trip and client size estimates.** (A) The average number of trips taken per guide and (B) the average number of clients per trip during each month of the study period accounting for data missing between January-April 2017. Error bars represent the standard error among individual guides.

we surveyed in this study to match that harvest estimate; however, our nighttime guided-gigging harvest estimate is likely an underestimate given photos were only available from May-December for that year. The nighttime guided recreational gigging harvest estimate will meet or exceed the total annual daytime private recreational harvest estimates for 2018 and 2019 if twelve additional nighttime guided-gigging charters (71% of identified charters) operate at the same average annual rate as the three charters we surveyed in this study.

## 4—Discussion

The Texas Southern Flounder population is facing a serious and prolonged decline and has not responded to increased regulations. We suggest this decline is likely due in part to uncaptured catch and effort occurring during the nighttime recreational gigging fishery. Managers have been hindered by data gaps from the nighttime guided gigging component of the recreational flounder fishery. Despite a limited sample size, this study highlights the efficiency of this component of the recreational flounder fishery and demonstrates that excluding estimates of

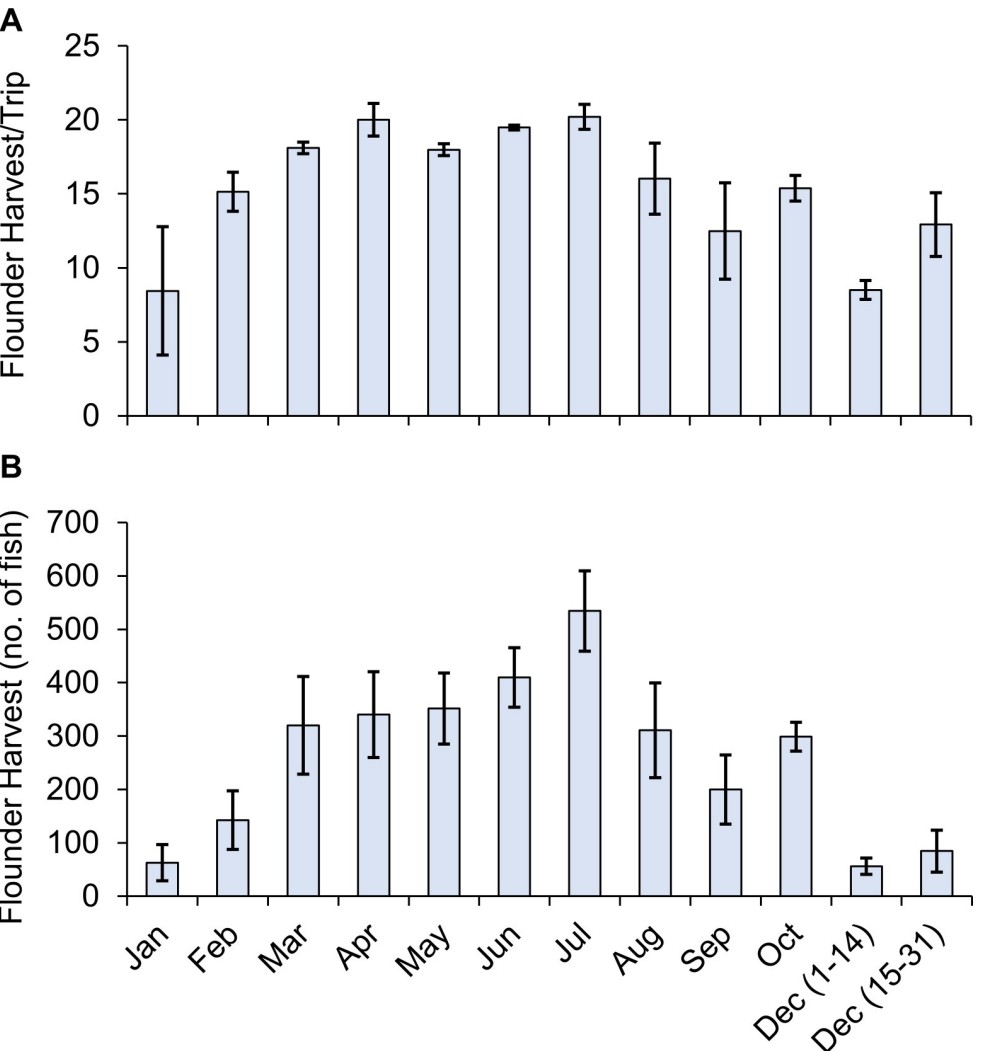

**Fig 3. Monthly trip and flounder harvest estimates.** (A) The average number of flounder harvested per trip and (B) the average total flounder harvest per guide during each month of the study period accounting for data missing between January-April 2017. Error bars represent the standard error among individual guides.

nighttime gigging harvest, particularly from guided-gigging charters, results in severely underestimating harvest patterns of Southern Flounder in Texas. Notably, this fishery's efficiency has dramatically increased with advances in modern recreational flounder gigging boat technology compared to historical wade-gigging. As a result, the disconnect between TPWD fishery-independent gill net data (long-term decline) and gigging guides' perceptions of the fishery (healthy) may be a byproduct of the extreme efficiency of this fishery and hyper-stable catches despite a declining population (e.g., [38]). Specifically, gigging guides may not observe declining catch rates and still reach their limit consistently because of the improved efficiency and creeping increase of their fishing power [39] that now allows a boat to cover miles in search of flounder over a relatively short time. Similarly, newer or younger guides may not perceive any changes because of shifting baselines [40]. We encourage that additional management resources be devoted to refining the quantifiable impacts that gigging boats (private and charter) have on Texas' recreational flounder fishery.

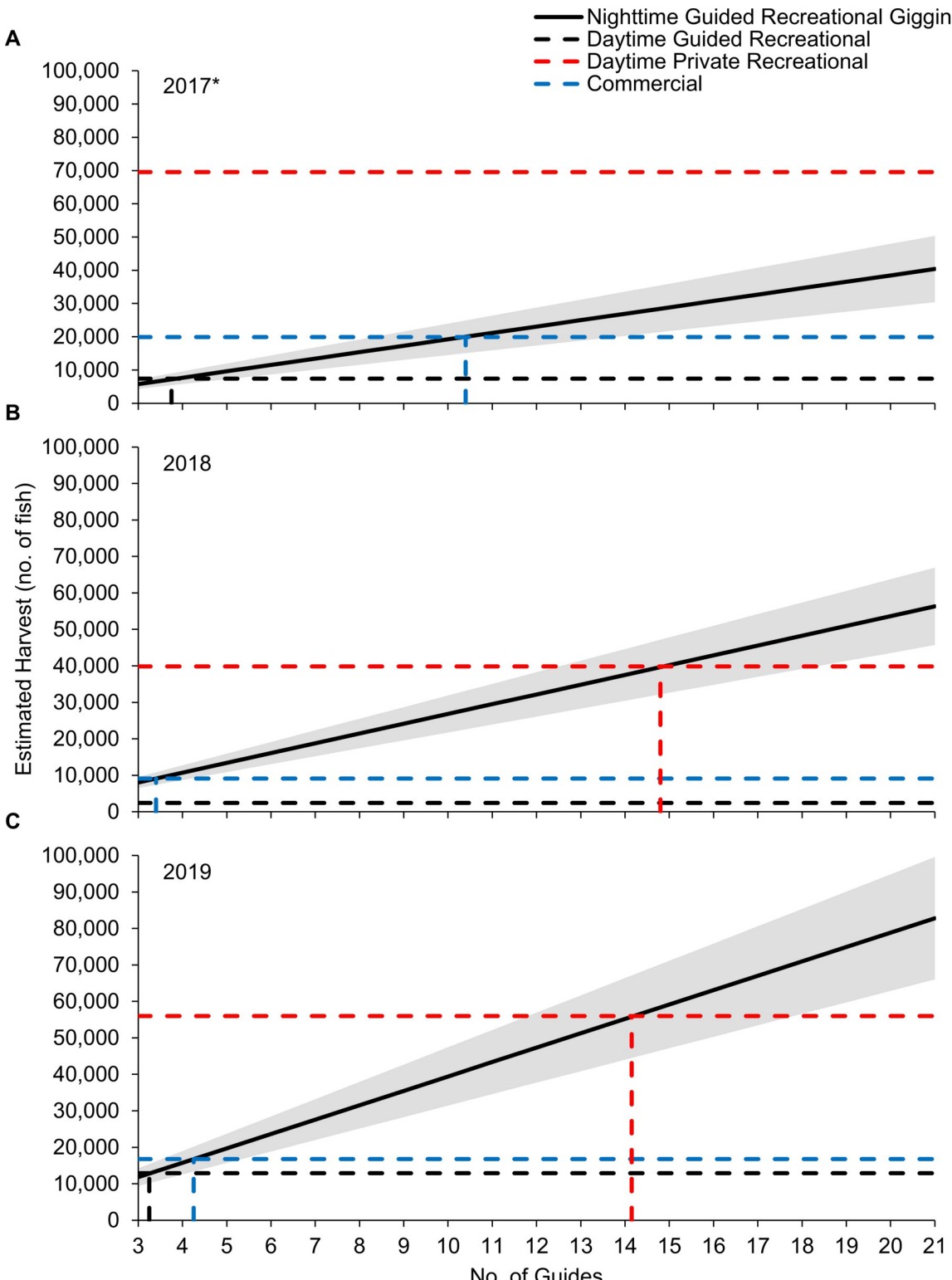

**Fig 4. Traditional statewide flounder harvest comparison.** Traditional statewide annual flounder harvest estimates for the daytime guided (black dashed line) and private (red dashed line) recreational fishery and commercial fishery (blue dashed line) compared to the nighttime guided recreational gigging harvest estimates (black solid line) generated in this study for (A) 2017, (B) 2018, and (C) 2019. Nighttime guided recreational gigging harvest was estimated across the range of identified flounder gigging guides operating in the state of Texas. Shaded areas represent the standard error associated with our estimates. Vertical dashed lines denote the intersection of

traditional statewide annual harvest estimates with the nighttime guided recreational gigging harvest estimates. *Note: data were not available for January-April of 2017; thus, nighttime guided recreational gigging harvest in 2017 is underestimated.

This study demonstrates how social media platforms may provide an open-source research tool to collect extensive data for harvest monitoring and aid the development of management priorities. Though, the information obtained from social media includes potential limitations and biases that need to be accounted for when utilizing such an approach [31]. For example, a limited number of guided-gigging charters regularly post on social media and intentionally exclude sharing spatial information on catch and effort. In addition, social media posts may be biased towards trips with higher catch rates, especially when posts are used for advertisement. In this case, the sampling bias would conservatively estimate overall trip and harvest metrics but potentially overestimate the average harvest per trip. Despite these limitations, these data revealed considerably large harvest estimates that are currently unaccounted for and further highlight the need for additional monitoring. For these reasons, pursuing a standardized quantification of recreational fishing effort and catches is of critical importance. Alongside traditional creel surveys, smart device applications have proven to be an effective and robust method for collecting data from recreational anglers to calculate more accurate harvest and effort estimates; though, these applications require high participation and user buy-in to be useful [25, 41].

Commercial fisheries have traditionally been identified as a primary driver of declining fish stocks globally; however, this study highlights how intense recreational fishing pressure, especially if inadequately monitored, can equally and even disproportionately contribute to lower fish stocks [42]. Based on the total trips surveyed in this study, results indicate that guided-gigging charters operate on over half the nights available to them during both the regular and restricted seasons and average approximately four clients per trip regardless of the month or harvest season. This consistent effort, combined with large client party sizes, and the fact that flounder may be harvested by gigging 335 days a year (321-day regular season, 14-day restricted season during the study period), indicates that guided-gigging charters have the capacity to exert extreme pressure on Texas' flounder population. The full extent of this pressure is difficult to calculate at a state level as no official numbers exist on how many guided flounder gigging businesses are currently being operated. Our preliminary internet search revealed that at least 21 licensed guides offer flounder gigging charters within the state; however, we believe this to be an underestimate. Many guided-gigging charters advertise on membership-only fishing forums, roadside signs, and boat ramp flyers, making their absolute number incredibly difficult to estimate. For this reason, requiring fishing guides licensed within the state to declare if they operate a flounder gigging charter service could benefit management.

The advanced capabilities of modern flounder gigging boats must also be considered. Contrary to traditional wade-gigging, gigging boats allow guides to quickly and easily service large client parties while maintaining high average harvest success rates. Across the study period, 95.2% of regular season and 98.6% of restricted season limits were filled each trip, with an average of 18.2 flounder being harvested per trip during the regular season. This average harvest per trip becomes even more important given that multiple Facebook posts within this survey contained information stating that guides conducted as many as three separate trips (different client parties on each trip) per night while only posting pictures from a single trip (i.e., once again indicating that our estimates are likely conservative). In comparison, commercial gigging operations can legally harvest a total of 30 flounder per night in Texas waters [43], suggesting it may become necessary to limit the number of trips any guided-gigging charter can

conduct within a 24-hour period. In addition, a large but unknown number of private anglers utilize the same highly efficient gigging boats for personal use. Currently, no data are available on the number of private gigging boats in Texas or on private nighttime gigging-boat harvest metrics. Obtaining data surrounding the private use of gigging boats is crucial to proper management. Despite management concerns, flounder gigging boats do fulfill an important role in the Texas recreational flounder fishery. They provide the public access to a culturally and economically important fishery and can accommodate small children as well as clients with mobility concerns. For these reasons, we encourage their continued use, albeit in a more monitored and regulated capacity.

Our results indicated that 61.3% of regular season and 76.6% of restricted season trips harvest ancillary species. However, the number of individual ancillary fish harvested nearly triples during the restricted season. This suggests that guided-gigging charters shift effort to target other species when flounder bag limits are reduced. Likewise, Sheepshead and Black Drum are the most common ancillary species to be harvested throughout the year. Given relatively large daily bag limits, (5 fish per person for each species), managers should consider that these species will likely become the primary focus for guided-gigging charters should flounder bag limits be reduced. Evidence of this shift is becoming apparent during Texas' new flounder season closure, (closed November 1st-December 14th), which was first enacted during 2021. For example, guided-gigging captains are already using social media to advertise charters specifically targeting ancillary species, mainly Sheepshead and Black Drum, during the new flounder closure, confirming our predictions. Some of these advertisement pictures depict full six-client limits of Sheepshead and claim that the charter operation has harvested over 650 Sheepshead and Black Drum during the six-week flounder closure with party limits on every trip. While managers are not currently concerned with Sheepshead and Black Drum populations, the ability of gigging operations to shift effort to these species must be accounted for in future management decisions. While these data here are too limited to make detailed conclusions, future studies should focus on characterizing the harvest of ancillary species on directed flounder trips.

Perhaps the most interesting portion of this study pertains to peaks in recreational guided-gigging effort and harvest occurring during the summer months, a time not traditionally associated with high flounder gigging effort. The few special surveys aimed at estimating Texas' recreational flounder effort were conducted in fall or winter months during the spawning migration, a period historically associated with peak nighttime recreational flounder harvest [14, 20, 26]. These studies likely missed the impacts of actual peak effort occurring during the summer months. The increase in summer guided-gigging pressure is most likely associated with summer tourism and the ability to cover large areas of water using gigging-boats, negating the need to wait for flounder to concentrate in tidal passes during the fall migration.

The discovery that recreational guided-gigging pressure and success is highest during the summer brings new considerations to the forefront of Texas' flounder management needs. This shift in recreational effort coincides with a historical shift in commercial gigging effort from November to spring in compliance with the November 30-day closure. Guided-gigging charters conduct the most trips and have extremely high success rates during months in which five fish per person per day can be harvested. The displacement of effort to spring and summer months are the unintended consequences of fall harvest and gear restrictions combined with highly efficient fishing technologies, thus demonstrating the adaptive behavior of fishers and creating new challenges to current management strategies [44, 45]. Given overharvesting concerns [11], a year-round reduction of the bag limit, while also keeping the new seasonal closure (i.e., zero flounder bag limit) from November 1st though December 14th (effective 2021), may be appropriate. In addition, data indicate that a smaller peak in effort and harvest levels occurs

each October. This is most likely due to the start of the spawning migration and angler anticipation of the November gigging closure. Despite this secondary peak, guided-gigging effort during September, October, and December was lower than they are during summer months, although party bag limit rates were a staggering 98.6%. Based on this evidence, the new annual complete flounder season closure (November 1st-December 14th; starting 2021) may slightly reduce annual guided-gigging harvest, especially of big old fat fecund female fish (BOFFFFs) which have a disproportionate effect on stock productivity and stability [46], while also limiting the economic impact on guided-gigging charters.

Traditional recreational flounder harvest metrics (both guided and private) are considerably lower than our estimate. These traditional recreational harvest estimates are generated using creel survey data that is collected during daytime hours and does not capture gigging effort as it is almost exclusively a nighttime activity. Given that nighttime gigging is the predominant method of recreational flounder harvest, the total annual flounder harvest is likely considerably underestimated. Our sensitivity analyses estimates indicate that for 2018 and 2019, statewide nighttime-guided gigging harvest likely exceeded the daytime-guided and commercial flounder fisheries. We suspect this was the case in 2017 as well, however only a partial year of data was available for analysis. It is worth noting that should all 21 identified gigging guides harvest at our estimated average annual rate, nighttime guided-gigging harvest would approach or exceed the combined total for all components of harvest currently estimated (i.e., TPWD daytime private recreational, TPWD daytime guided recreational, and TPWD commercial). We also note that our estimates only applied to guided recreational gigging. Harvest and effort for the private recreational gigging component remains unknown but is likely high given the large number of participants and high gear efficiency (via private gigging boats). The considerably large nighttime flounder harvest and effort that is currently unaccounted for demonstrates a clear need for additional monitoring and development of novel reporting systems for all components of Texas' recreational flounder fishery, particularly the nighttime guided and private recreational components as they likely account for the majority of the total annual flounder harvest [20]. It is critical that accurate and current harvest and effort metrics are made available to allow managers to better evaluate existing regulations and ensure a sustainable fishery.

## 5—Conclusions

Guided-gigging charters appear capable of placing heavy, year-round pressure on the Texas flounder population. Additional information pertaining to statewide nighttime effort, recreational gigging-boat use (both private and guided), and summer harvest, must also be obtained and factored into future management decisions. This study demonstrated how data gathered from social media can greatly inform scientific research and reveal conservation concerns highlighting the need for improved monitoring and management strategies. This methodological approach may be opportunistically applied in other regions worldwide; however, due consideration must be given to potential inherent biases when using social media platforms for data collection, analysis, and interpretation [31]. Ultimately, this study indicates that guided-gigging charter businesses operate within the law and are observant of recreational harvest limits. Thus, we are confident that constructive dialogue between managers and gigging charter captains could lead to effective management changes and a more sustainable recreational flounder fishery in Texas.

## Author Contributions

**Conceptualization:** Quentin A. Hall, Gregory W. Stunz.

**Data curation:** Quentin A. Hall, Daniel M. Coffey, Matthew K. Streich.

**Formal analysis:** Quentin A. Hall, Daniel M. Coffey, Matthew K. Streich.

**Investigation:** Quentin A. Hall, Daniel M. Coffey, Matthew K. Streich, Gregory W. Stunz.

**Methodology:** Quentin A. Hall.

**Project administration:** Gregory W. Stunz.

**Resources:** Mark R. Fisher.

**Supervision:** Matthew K. Streich, Gregory W. Stunz.

**Validation:** Mark R. Fisher.

**Writing – original draft:** Quentin A. Hall, Daniel M. Coffey, Matthew K. Streich.

**Writing – review & editing:** Quentin A. Hall, Daniel M. Coffey, Matthew K. Streich, Mark R. Fisher, Gregory W. Stunz.

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
