## [Decision Letter · Decision Letter 0]

21 Feb 2022

PONE-D-21-39734Social media shines light on the “hidden” impact of nighttime guided-gigging charters on Texas’ Southern Flounder fishery: A stab in the dark.PLOS ONE

Dear Dr. Hall,

Thank you for submitting your manuscript to PLOS ONE. After careful consideration, we feel that it has merit but does not fully meet PLOS ONE’s publication criteria as it currently stands. Therefore, we invite you to submit a revised version of the manuscript that addresses the points raised during the review process.

We look forward to receiving your revised manuscript.

Kind regards,

Charles William Martin

Academic Editor

PLOS ONE

Journal Requirements:

2. PLOS ONE requires that the research must be described in enough detail to allow readers to fully replicate the study (https://journals.plos.org/plosone/s/criteria-for-publication#loc-3), and that all data underlying the findings described in their manuscript must be fully available (https://journals.plos.org/plosone/s/criteria-for-publication#loc-7).

Please could you provide the links to the collected material from Facebook and a statement indicating that the use of this dataset was done in compliance to the Facebook Terms and Conditions and our requirements for this type of study (https://journals.plos.org/plosone/s/submission-guidelines#loc-personal-data-from-third-party-sources).

Reviewers' comments:

Reviewer's Responses to Questions

**Comments to the Author**

1. Is the manuscript technically sound, and do the data support the conclusions?

Reviewer #1: Yes

Reviewer #2: Partly

Reviewer #3: Yes

2. Has the statistical analysis been performed appropriately and rigorously? 

Reviewer #1: Yes

Reviewer #2: Yes

Reviewer #3: Yes

3. Have the authors made all data underlying the findings in their manuscript fully available?

Reviewer #1: Yes

Reviewer #2: No

Reviewer #3: Yes

4. Is the manuscript presented in an intelligible fashion and written in standard English?

Reviewer #1: Yes

Reviewer #2: Yes

Reviewer #3: Yes

5. Review Comments to the Author

Reviewer #1: This paper addresses a problem that is vitally important to the future management of an important commercial and recreational species that has continued declining across its entire range despite revised management efforts. Fisheries scientists are learning that environmental factors and climate change could potentially explain some of the continued declines. The authors did a good job of addressing and explaining this problem with relevant citations and bring to light a new problem that myself, and I'm sure many other flounder researchers, have pondered for some time. Unreported recreational and private gigging effort is something that is logistically difficult to quantify, thus has not been accurately described in yearly harvest estimates. The authors of this study use novel methods to leverage social media as a tool for quantifying overall effort and catch efficiency of the private for-hire gigging sector of the Texas fishery. The quantified estimates of effort, including number of gigging trips per guide per year (~153), harvest per trip (~18 fish), trip limit efficiency (95%), and annual number of fish harvested per guide (2,766) are important metrics for managers to consider. Many of the points made in paper should be considered in both immediate management and serve to highlight one potential direction of the "next generation" of Southern Flounder research.

The only recommendation I have would be to elaborate further on the assumptions and potential bias of the total nighttime guided recreational gigging harvest estimates. The final estimate of total harvest assumes that the effort and efficiency derived from 3 gigging guides that post the most pictures on social media is evenly distributed across the other 21 guides. Often times the people that post the most are the best at what they do, but this is not always the case. It also assumes that there is sufficient clientele to fill those trips, which, at an average of 4 people per trip x 153 trips per year x 21 guides would be nearly 13 thousand clients per year taking flounder gigging charters in Texas. Again, not unreasonable but the authors should consider the likelihood that this demand exists for flounder gigging charters in Texas. And if it does, or demand is exceeded, it would give a lot of support to the final estimates. Another point that could be made is that the estimates of total private for-hire gigging harvest presented in this paper could actually be conservative when it comes to the overall gigging harvest given that private recreational gigging effort has not been effectively quantified.

In my opinion, this is the only consideration/edit I would recommend. The authors did a good job of acknowledging the assumptions made in other areas of the paper; however, I think it is important to acknowledge the assumption that effort and efficiency is evenly distributed across the rest of the for-hire gigging fleet makes the total nighttime guided recreational gigging harvest estimate on the upper end of what is possible from a fully operational and fully efficient fleet of 21 gigging charter guides.

I also agree with the authors recommendations for obtaining more data from recreational and private gigging; specifically, requiring fishing guides licensed within the state to declare if they operate a flounder gigging charter service. Another way would be mandatory catch reporting similar to MRIP or Snapper Check used in the red snapper fishery. More data on nighttime gigging harvest for areas in the northern Gulf of Mexico is going to be of the upmost concern in the coming years if limit changes, hatchery efforts and spawning season closures implemented in the past couple years are not positively reflected in upcoming stock assessments.

Reviewer #2: I thank the authors for the opportunity to review "Social media shines light on the “hidden” impact of nighttime guided-gigging charters on Texas’ Southern Flounder fishery: A stab in the dark." Overall, I found the manuscript interesting and very likely to be identifying a missing source of fishing mortality in the Texas Southern Flounder fishery. Overall, I think this study could be publishable; however, I have one major comment that gives me pause about the results and interpretation.

Major Comment:

The only major issue I came across while reading this manuscript had to do with some of the assumptions in the methods. Specifically, the authors found the three of the most active gigging operations in the "high effort geographic area" of Texas, and then extrapolated harvest from this small, high-avidity sample size across the entire fishery. As I read it, the estimates that authors come up with are possible, but also represent the upper end of the range of estimates. In this regard, I think the manuscript needs to quantify some of the assumptions that were made. For example, it is plausible that the three guides that were studied were the most active in the guide fishery (at least the authors present this possibility)—so what if other gig guides are not that active? What about if other gig guides don't fish as many days of the year? Or they don't approach the creel limit as often? There are several possible factors that could result in a more heterogeneous gig-guide fishery, and exploring those factors would almost certainly result in a lower estimate than the authors arrived at. I'm not particularly set in one way or another to include this variability. Some of it could be done earlier in the methods if an "avidity" metric could be developed for guides and then applied across the population of guides? Another way could be to stick with the current method and call it an upper estimate. Then, explore what the estimate(s) might be if part of the guide fishery fewer days, and/or creeled fewer fish. What I have in mind is a sort of sensitivity analysis that could explore realistic lower estimates. I don't know what could be called the low-end estimate, but I also don't think the low or high estimate is the story—I think if the range of plausible estimates is still a significant missing piece of the southern flounder fishery, then that is the story. (Or, if a low-end estimate is relatively small, then maybe the gig-guide fishery is not as extractive as thought? Just trying to consider all possible interpretations.)

One smaller issue is that the data are not really available for authors to access. I understand where the authors are coming from by saying that the data are available online, because they mined Facebook. However, as I understand it a reader would need to replicate their study (which is currently not possible; see comments elsewhere), in addition to the fact that the raw data is stored by Facebook, which is a proprietary website in which users can add and remove data. For example, if the gig-guides remove photos or removed their Facebook account, this data would never be available. Perhaps what the authors have done is acceptable based on the PLoS data policy, but in the spirit of open data, this manuscript does not have data available for readers. I don't think the authors need to provide all the actual photos they analyzed, but I assume there is a spreadsheet of data that was analyzed and which could be made available?

Minor comments:

Why is there a period in the title?

L28: How do you know gigging activity is high, if the first part of the sentence establishes that this activity is not accounted for?

L72: Why quotes on push fans? Does the term mean something different without quotes?

L78: Can you please cite the claim that gig charters perceive a healthy flounder population?

L100: I'm not against the Froeschke et al. citation, but I wonder if it should be caveated that the data used in that paper ended in 2008—nearly 15 years ago. And while I don't necessarily question that study, the flounder declines of the last ~5 years do appear to be related to recruitment. I think my larger point here is to consider that fact that Texas (I think) is dealing with possibly two flounder declines: 1) a chronic adult decline, and 2) a recent and acute recruitment decline. I mention this because they have the possibility to be quite different, ranging from their mechanisms to their population-level effects.

L165–66: Can you please provide more details about the internet search (in theory, so that the methods are reproducible)? For example, were all the terms include in a search? Only one term at a time? Combinations of terms (if so, which combinations and why?) What dates were the searches performed (this is all routine methods for internet searches.) Also, can you please provide specifics on fishing forms, outdoor classified boards, etc. I could imagine a table of URLs as a supplement. I recognize this is a bit detailed, but if this is the primary means of data generation, I think it should be iron-clad in terms of what was done so readers know where exactly this data came from and how they could reproduce it. Right now a reader could not replicate your methods.

L169: How were these three subset? Why only three? How many in total were there? (You mention this was based on extensive photos, but then please quantify extensive.)

L175-76: Just a few lines above you state "...charters operate....high effort geographic area..." Yet here you assume the charters to be representative of average guide-gigging. This seems conflicted and likely to overestimate gigging catch and effort because you have biased the sample to the highest effort areas and the guides that post the most (based on your description). [See major comment above.]

L183: How did you check for duplicate photos? For example, it seems realistic that on a poor night of gigging, and operating service might re-post a photo from a more productive night as there is a financial incentive not to post the actual catch if it is low.

L196: This seems like a safe assumption, but one you could also check. For example, look back at weather data for storms, precipitation, and wind events that would have precluded gigging. If no gigging photos were posted on those days, your assumption is strengthened. If guides posted gigging photos on those days, that is a problem and the assumption does not hold. I'm not sure what you would do if guides are posting trip photos on days it would have been impossible to fish.

L197: What is the basis for the assumption that each guide posted photos of each trip? I don't think this is unreasonable, but would be nice to pin that assumption on something.

L254 and elsewhere: Why standard error and not standard deviation? I'm not suggesting which is right or wrong, but SD would be easier to interpret for some of the estimates you report.

L269-72: In relation to previous comments, I am concerned that the gigging estimates are biased, and therefore this multiplication factor would be applying the highest catch estimates across all gigging operations, which are likely to operate at very different levels. [See major comment above.]

L329-331: What is meant by high level of consistency? My first impression of these small standard errors is that they are smaller because the sample size for these months is larger, and with the same distribution of data (and variance), SE will decrease with sample size. I'm sure the authors know this, but perhaps my hang up is not knowing what "consistency" means here or how that translates to the statistical evidence.

L379-80: Are the authors suggesting that the entirety of the flounder decline is due to missing nighttime gigging catch? While I agree that the authors make a compelling case that gig data is missing from this fishery, I think it is an overreach to cite it as the sole factor? Furthermore, flounder have been declining throughout their range, including many places where gigging is not expected to be taking place or at least playing the role it is in Texas.

I have no real issues with the Discussion, because I think it appropriately interprets the current results. However, based on my major comment above, I think the methods and results need revision to account for potential biases in harvest (over)estimates. Revisiting the methods and results to account for bias may well change the Discussion, and so I will withhold any major comments on the discussion.

Just a few observations on the figures: Figure 1 is not colorblind-friendly. Figures 2 and 3 (and probably 4) would be easier to interpret as a box plot or something that better captures the distributional nature of the data (see Newman and Scholl 2012). For Figure 4, I would also suggest changing from a bar plot to box plot, but minimally bars are easier to interpret when not stacked (rather, clustered so they all originate at y = 0). Stacked bars do not have common reference locations, making comparison difficult. Also, what does the asterisk represent? I see a note in the Figure caption, but I'm still not 100% sure. (Also, the symbols do not match.)

Newman, G.E. and Scholl, B.J., 2012. Bar graphs depicting averages are perceptually misinterpreted: The within-the-bar bias. Psychonomic bulletin & review, 19(4), pp.601-607.

Reviewer #3: This manuscript uses social media to inform critical data gaps in the management and regulation of the southern flounder fishery in Texas. The methodology employed in this manuscript is a novel (especially for fisheries applications) and creative solution to a complex problem. Despite several acknowledged caveats, the manuscript makes a strong case for the nighttime guided gigging having an appreciable impact on the sustainability of the fishery. At a minimum, the results presented in the manuscript highlight the need for expanded collection and inclusion of the nighttime gigging harvest in management practices. The manuscript is well written, conceptually solid, and not only likely to impact the management of Texas’ southern flounder fishery but also has the potential to influence the management of wildlife and fisheries elsewhere. I recommend this manuscript for publication with minor revisions.

Major concern:

The authors detail several potential biases in their methodology and ultimately conclude that their results underestimate the effort and harvest. While I agree with their conclusions as it pertains to these specific biases, I am concerned that their method for selecting the representative guide operations has introduced an overlooked bias. The three guide operations that serve as the basis for calculating multiple metrics were selected because each had an extensive social media presence (LINE XXXX). We know that social media influences consumer behavior and brand attitudes (REF) and if recreational fishermen use the same approach for selecting a guided excursion, then the authors may have selected the three most popular guide operations. In that case, extrapolating the metrics from these operations to the other 21 (18?) guide operations could lead to an overestimation of effort and harvest. How large of an overestimation is unknowable but it warrants some text in the discussion. As a suggestion only, an interesting way to summarize the uncertainties and potential biases would be to include a small table that outlines the risks to interpretation of the data and management actions associated with each bias.

6. PLOS authors have the option to publish the peer review history of their article (what does this mean?). If published, this will include your full peer review and any attached files.

Reviewer #1: **Yes: **Dylan M. Kiene

Reviewer #2: No

Reviewer #3: **Yes: **Michael Lowe

---

## [Author Response · Author response to Decision Letter 0]

19 Apr 2022

Please refer to response to Editor and reviewers document: (Response to Reviewers.docx).

---

## [Editor Report · Decision Letter 1]

20 May 2022

Social media shines light on the “hidden” impact of nighttime guided-gigging charters on Texas’ Southern Flounder fishery: A stab in the dark

PONE-D-21-39734R1

Dear Dr. Hall,

We’re pleased to inform you that your manuscript has been judged scientifically suitable for publication and will be formally accepted for publication once it meets all outstanding technical requirements.

Kind regards,

Charles William Martin

Academic Editor

PLOS ONE

Additional Editor Comments (optional): The authors have done a nice job with the revision and taking reviewer comments into account.
---

## [Editor Report · Acceptance letter]

25 May 2022

PONE-D-21-39734R1 

Social media shines light on the “hidden” impact of nighttime guided-gigging charters on Texas’ Southern Flounder fishery: A stab in the dark 

Dear Dr. Hall:

I'm pleased to inform you that your manuscript has been deemed suitable for publication in PLOS ONE. Congratulations! Your manuscript is now with our production department. 

Kind regards, 

on behalf of

Dr. Charles William Martin 

Academic Editor

PLOS ONE